# DMQR-RAG: Diverse Multi-Query Rewriting in Retrieval-Augmented Generation

## Abstract

Large language models often encounter challenges with static knowledge and hallucinations, which undermine their reliability. Retrieval-augmented generation (RAG) mitigates these issues by incorporating external information. However, user queries frequently contain noise and intent deviations, necessitating query rewriting to improve the relevance of retrieved documents. In this paper, we introduce DMQR-RAG, a Diverse Multi-Query Rewriting framework designed to improve the performance of both document retrieval and final responses in RAG. Specifically, we investigate how queries with varying information quantities can retrieve a diverse array of documents, presenting four rewriting strategies that operate at different levels of information to enhance the performance of baseline approaches. Additionally, we propose an adaptive strategy selection method that minimizes the number of rewrites while optimizing overall performance. Our methods have been rigorously validated through extensive experiments conducted in both academic and industry settings.

## 1 Introduction

Large language models (LLMs) possess powerful comprehension and generation abilities (Touvron et al., 2023a;b), demonstrating remarkable performance across various downstream tasks (Jiang et al., 2023; Su et al., 2024). However, the parametric knowledge within LLMs is inherently static, making it challenging for them to provide up-to-data information in real-time scenarios (Yao et al., 2023). Additionally, LLMs are prone to hallucinations when addressing factual questions (Guan et al., 2024; Hoshi et al., 2023), undermining the reliability of generated answers.

To address these issues, retrieval-augmented generation (RAG) (Gao et al., 2023b; Chen et al., 2024) has emerged as a method to enhance LLMs by retrieving and incorporating external knowledge. However, due to noise and intent bias in the original queries, direct retrieval often fails to yield sufficiently relevant documents (Ma et al., 2023; Chan et al., 2024). Therefore, query rewriting is critical for retrieving the pertinent documents as shown in Figure 1(b).

Substantial research has explored improved methods for query rewriting (Chan et al., 2024; Wang et al., 2024a; Mao et al., 2024; Wang et al., 2023a; Zheng et al., 2024; Ma et al., 2023), which can be broadly categorized into two families: training-based and prompt-based approaches. Training-based methods (Wang et al., 2024a; Mao et al., 2024; Ma et al., 2023) involve supervised fine-tuning of models using annotated data or reinforcement learning, leveraging downstream retrieval metrics as rewards. In contrast, prompt-based methods (Zheng et al., 2024; Chan et al., 2024; Wang et al., 2023a) utilize prompt engineering to guide LLMs in specific rewriting strategies. However, most methods generate a single rewritten query, which often leads to a lack of diversity in the retrieved documents. This results in a low recall of genuinely relevant documents, as demonstrated by our experiments. Furthermore, some methods focus on specific query types, such as complex multi-hop or multi-intent queries, limiting their applicability for general-purpose queries [1].

---

[1] For example, "Where are the authors of the Transformer paper currently working?" is a multi-hop query, while "Which paper has more citations, the Transformer paper or the ResNet paper?" is a multi-intent query. Both require multiple retrievals to answer. In contrast, "What is the citation count for the Transformer paper?" is a general-purpose query that can be answered with a single retrieval.

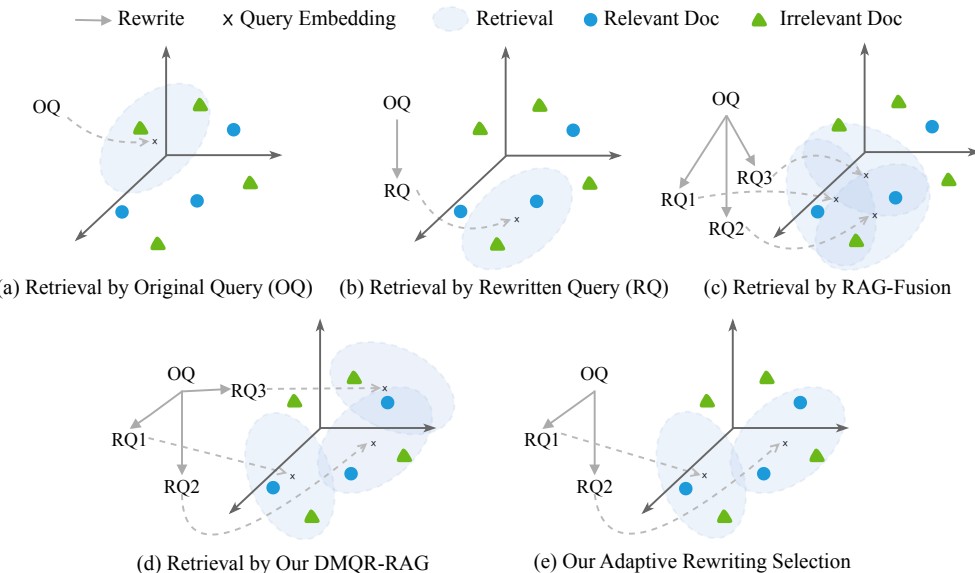

Figure 1: The motivation of our work. (a) Users often struggle to express their intentions accurately, which can lead to the retrieval of irrelevant documents. (b) In some cases, rewritten queries can successfully retrieve relevant documents. (c) Rewritten queries that are similar (i.e., lacking diversity) may yield similar document retrievals, potentially overlooking other relevant documents. (d) Our DMQR-RAG encourages diverse rewritten queries, resulting in a broader range of retrieved documents that encompass all relevant items. (e) Our adaptive rewriting selection eliminates unnecessary rewrites without compromising relevant document retrieval, while also reducing noise by minimizing the retrieval of irrelevant documents.

In this paper, we propose DMQR-RAG, *a general-purpose multi-query rewriting method* aimed at retrieving diverse documents with a high recall of relevant documents in RAG. Our motivation is shown in Figure 1. The most relevant approach to our work is RAG-Fusion (Rackauckas, 2024), which utilizes multiple rewriting queries to retrieve additional documents and applies the Reciprocal Rank Fusion algorithm (Cormack et al., 2009) for reranking. Unlike RAG-Fusion, our method is inspired by information diversity to enhance the transmission of information from queries to documents (Maron & Kuhns, 1960; Baeza-Yates et al., 1999; Weikum & Vossen, 2001), leading to the development of four rewriting strategies based on different levels of information. Each rewritten query can retrieve different documents, and we subsequently use a cross-attention embedding model to rerank these documents. However, increasing the number of rewritten queries is not always beneficial, as it can introduce noise. To address this, we employ a rewriting strategy selection method that adaptively identifies a limited number of suitable strategies for rewriting and retrieval based on the specific query. Our contributions are summarized as follows:

- We propose a general-purpose multi-query rewriting framework that employs various strategies based on the amount of information. Our findings indicate that multi-query rewriting generally outperforms single-query rewriting, with our information-based multi-query approach often surpassing vanilla RAG-Fusion.

- We introduce a rewriting strategy selection method that identifies the most suitable approach for each query, achieving better performance in both document retrieval and final responses in RAG with fewer rewrites.

- To facilitate fair comparisons of the rewriting module in RAG, we establish a standardized setup for rewriting based on well-established practices, mitigating the influence of extraneous variables within the complex RAG pipeline. Extensive evaluations under both academic and industry settings validate the effectiveness of our methods.

## 2 METHODOLOGY

We first introduce the traditional RAG workflow and propose a standardized setup to explore how the rewriting strategy impacts performance. Building on this foundation, we present our DMQR-RAG framework and the adaptive rewriting selection method.

### 2.1 FORMAL SETUP FOR MULTI-QUERY REWRITING

Given a user's query $q$, the traditional RAG process begins by rewriting the query to obtain $q'$. Next, a retriever searches for relevant documents, represented by the set $D$. These documents are then reranked to produce $D'$. Subsequently, the top $K$ documents are concatenated with the original query $q$ and input into an LLM to generate the final response $A$. The entire process is as follows:

$$q' = \text{Rewriter}(q), \quad D = \text{Retriever}(q'), \quad D' = \text{Reranker}(D), \quad A = \text{LLM}(q, \text{TopK}(D')). \quad (1)$$

However, the elongated pipeline introduces multiple processing steps, each of which can be executed in various ways, impacting the final outcome (Wang et al., 2024b). These factors include, but are not limited to: the source and quality of documents, the chunking strategy, the embedding model for document chunks, the vector database for storing embeddings, the algorithms used for retrieval (keyword, semantic, or hybrid), and the reranker and LLM models. In this context of high variability and complexity, it is challenging to independently evaluate the effectiveness of the rewriting module. Therefore, a formal setup tailored for rewriting is essential.

Without loss of generality, we propose to standardize the Retriever and Reranker to current mainstream methods, and evaluate the (multi-query) Rewriter using different LLMs to demonstrate that the rewriting approach is generalizable across various LLMs in a recognized and fair context. Specifically, we treat the Retriever as a black box, using results from the Bing search engine to ensure data timeliness and to avoid the complexities of hyperparameter tuning required by other retrieval methods. Additionally, we employ the widely adapted BAAI-BGE-reranker (Xiao et al., 2023) as the Reranker. In summary, our formal setup for multi-query rewriting can be defined as follows:

$$\boldsymbol{q'} = \{q, \text{RS}_1(q), \text{RS}_2(q), ..., \text{RS}_n(q)\}, \quad (2)$$

$$D = \text{Retriever}_{Bing}(\boldsymbol{q'}), \quad D' = \text{Reranker}_{BGE}(D), \quad A = \text{LLM}(q, \text{TopK}(D')). \quad (3)$$

where $\text{RS}_i$ denotes different rewriting strategies, and $\boldsymbol{q'}$ contains the original query $q$ along with all its rewritten versions [2]. These queries are used for retrieval, and all the retrieved documents are collated and submitted to the downstream modules.

### 2.2 THE DIVERSE MULTI-QUERY REWRITING FRAMEWORK

Due to their advanced natural language understanding capabilities (Touvron et al., 2023a;b), LLMs are often used as the foundational tool for query rewriting. In this section, we will first explore various LLM-based rewriting strategies from an informational perspective, followed by an introduction to the rewriting strategy selection method.

#### 2.2.1 REWRITING STRATEGIES

Current rewriting methods often rely on a straightforward, single rewrite of the original query, which frequently fails to retrieve relevant documents. Additionally, multi-query rewriting approaches, such as RAG-Fusion (Rackauckas, 2024), tend to be simplistic, producing rewrites that are nearly identical and lacking in diversity. This limitation hampers their ability to enhance overall performance.

An effective multi-query rewriting strategy should meet the following informational criteria: *each rewritten query must be diverse, providing unique information not present in the others.* By enhancing the diversity of information in the rewritten queries, we increase the likelihood of retrieving a broader range of documents, ultimately improving our chances of obtaining genuinely relevant documents (Maron & Kuhns, 1960; Baeza-Yates et al., 1999; Weikum & Vossen, 2001).

---

[2]Note that incorporating the original query $q$ is essential because some users can indeed express their intentions accurately, providing valuable context that improves the relevance of the retrieved documents.

Specifically, we propose four rewriting strategies that focus on adjusting or preserving the information in the original query. These strategies aim to refine the query while also providing unique insights to facilitate the retrieval of a diverse set of documents.

**Information Equality.** User-generated queries often contain irrelevant noise and unclear intent (Gao et al., 2023b), leading to deviations from the intended retrieval objective. Therefore, a general rewriting approach is necessary to denoise and refine the original query. We refer to this method as General Query Rewriting (GQR), which refines the original query $q$ while retaining all relevant information and eliminating noise, thereby enhancing retrieval precision. This strategy is in line with the approach proposed by Ma et al. (2023).

Furthermore, to ensure alignment with search engine preferences while maintaining the same amount of information, we introduce Keyword Rewriting (KWR). This strategy aims to extract all keywords from the query $q$, particularly nouns and subjects. By doing so, KWR enables search engines to directly address user needs and quickly locate relevant documents (Gupta & Vidyapeeth, 2017), while also reducing the parsing burden on the search engine.

**Information Expansion.** By incorporating prior information into the original query, we can assist the retriever in recalling a more diverse range of documents for responses (Gao et al., 2023a; Wang et al., 2023a). We propose leveraging the prior knowledge of LLMs to generate a pseudo-answer for retrieval, a method we term Pseudo-Answer Rewriting (PAR). The rationale behind this method is twofold: first, inherent semantic differences between queries and answers can introduce retrieval biases; second, the pseudo-answer enriches the original query with additional information. Although this pseudo-answer may not be factually accurate, it is semantically aligned with the real answer and can capture relevant response patterns, aiding in the retrieval of more pertinent documents, especially when LLMs encounter hallucinations (Guan et al., 2024; Hoshi et al., 2023).

**Information Reduction.** When a query contains excessive detail, the retriever often struggles to identify the most essential and useful information (Zheng et al., 2024), resulting in discrepancies between the retrieved documents and the user's primary needs. Additionally, this situation places a significant burden on downstream modules, such as the reranker and generator (Wang et al., 2023b). Therefore, it becomes crucial to discard superfluous details and extract key information. We define this strategy as Core Content Extraction (CCE).

To maintain universality, we combine all strategies to create a scalable strategy pool, denoted by

$$\mathcal{RS} = \{\text{RS}^{\text{GQR}}, \text{RS}^{\text{KWR}}, \text{RS}^{\text{PAR}}, \text{RS}^{\text{CCE}}, \dots\}, \tag{4}$$

where $\text{RS}^{\text{GQR}}, \text{RS}^{\text{KWR}}, \text{RS}^{\text{PAR}}, \text{RS}^{\text{CCE}}$ represent the four rewriting strategies mentioned above [3]. This strategy pool can dynamically incorporate new rewriting strategies based on practical needs, ensuring that our method remains flexible and universally applicable. For example, we can include sub-query rewriting for multi-hop queries and even integrate training-based methods to enhance effectiveness further (Wang et al., 2024a; Mao et al., 2024; Ma et al., 2023).

### 2.2.2 Adaptive Rewriting Strategy Selection

In practical industrial scenarios, user queries are diverse. While multi-query rewriting can enhance retrieval diversity, applying a fixed set of strategies to every query is not optimal. Therefore, it is crucial to dynamically select rewriting strategies tailored to each specific query, generating multiple rewritten queries that best suit the original intent.

We implement this selection method using lightweight prompting and few-shot learning. Specifically, we incorporate descriptions of the rewriting strategies in the strategy pool $\mathcal{RS}$ into the prompts of the LLMs as contextual information. These descriptions outline the applicable query types and the roles of each strategy, enabling the LLMs to gain a comprehensive understanding of all available rewriting strategies. To enhance strategy selection in challenging cases, we also adopt a few-shot approach by providing the LLMs with multiple demonstrations, which assist them in selecting suitable rewriting strategies for difficult queries. The detailed prompt is shown in Table 1.

---

[3]The details of the prompts used for these rewriting strategies with LLMs are provided in the Appendix.

Table 1: The prompt for adaptive rewriting strategy selection.

---

### Instruction ###
You will receive a user's question that requires retrieving relevant content through internet search to provide an answer. There are now the following `{N}` rewriting methods, `{RS name list}`. Based on the characteristics of the query, please select some of the rewriting methods to rewrite the question.

### $\{RS_i$ name$\}$ ###
`{`$RS_i$ `description}`
...

### Guidelines ###
`{`$RS_i$ `name}`: `{`$RS_i$ `usage guideline}`
...

### Output Format ###
Each output line should list the selected rewriting method, starting with its name followed by the rewritten result.
The final line should explain the selection rationale for these methods and the exclusion of others, beginning with "reason: ".

### Example ###
Question: `{example query}`
Output:
`{`$RS_j$ `name}`: `{Rewriting result based on RS`$_j$`}`
`{`$RS_k$ `name}`: `{Rewriting result based on RS`$_k$`}`
reason: `{the rationale for selecting these methods over others}`

Begin! Only output the final result without any additional content. Do not generate any other unrelated content.
Question: `{query}`
Output:

---

## 3 EXPERIMENTS

### 3.1 EXPERIMENTAL SETUP

#### 3.1.1 DATASETS

We conduct experiments using three representative open-domain question-answering datasets: (1) AmbigNQ (Min et al., 2020), designed to address the inherent ambiguity in Natural Questions (Kwiatkowski et al., 2019); (2) HotpotQA (Yang et al., 2018), which includes complex questions requiring multi-hop reasoning, with the validation set used for evaluation due to the lack of ground truth in the test set; and (3) FreshQA (Vu et al., 2024), a dynamic benchmark that encompasses various question types and necessitates up-to-date world knowledge for accurate responses. We also conduct experiments on industry datasets, which will be described later.

#### 3.1.2 METRICS

We evaluate both retrieval and end-to-end response metrics. Specifically, we assess rewriting effectiveness using the Top-5 hit rate (H@5) and precision (P@5) of the retrieved documents, with relevance evaluated by GPT-4. For end-to-end responses, we use official evaluation methods: for HotpotQA and AmbigNQ, we calculate exact match (EM) and F1 scores, while for FreshQA, we use GPT-4 to score responses and compute accuracy (Acc).

Table 2: The results of using different rewriting methods on three representative datasets are presented. Best result is in boldface, and the second best is underlined. Due to the iterative query rewriting and retrieval approach employed by RQ-RAG, we cannot directly assess the quality of retrieval; therefore, only end-to-end results are provided here.

| Method | AmbigNQ | | | | HotpotQA | | | | FreshQA | | |
|---|---|---|---|---|---|---|---|---|---|---|---|
| | H@5 | P@5 | EM | F1 | H@5 | P@5 | EM | F1 | H@5 | P@5 | Acc |
| OQR | 80.04 | 47.02 | 51.28 | 63.47 | 42.24 | 18.99 | 35.81 | 47.83 | 71.50 | 45.57 | 69.00 |
| *Finetuning-based, Single-query rewriting* | | | | | | | | | | | |
| RRR | 73.47 | 44.78 | 49.28 | 62.17 | 38.39 | 18.28 | 36.49 | 48.05 | 65.67 | 40.53 | 66.50 |
| RQ-RAG | - | - | 52.48 | 63.96 | - | - | 40.32 | **54.11** | - | - | 69.67 |
| *Prompt-based, Single-query rewriting with GPT-4* | | | | | | | | | | | |
| Rewrite | 81.06 | 47.17 | 51.24 | 63.96 | 42.08 | 18.91 | 35.47 | 47.39 | 70.67 | 44.61 | 70.83 |
| Hyde | 79.65 | 59.36 | 53.95 | 64.83 | 46.92 | 25.23 | 39.36 | 51.74 | 61.10 | 44.04 | 62.83 |
| *Prompt-based, Multi-query rewriting with GPT-4* | | | | | | | | | | | |
| RAG-Fusion | 86.33 | 53.62 | **55.47** | **68.59** | 51.58 | 26.91 | 40.00 | 53.56 | 76.17 | 60.00 | 74.50 |
| DMQR-RAG (ours) | **88.08** | **62.43** | 55.24 | 68.57 | **54.18** | **27.93** | **41.12** | 53.99 | **77.83** | **60.03** | **76.67** |

### 3.1.3 BASELINES

We adopt prompting and fine-tuning methods as our baseline approaches. For prompt-based methods: (1) LLM Rewrite (abbreviated as Rewrite) (Ma et al., 2023) utilizes prompts to harness the inherent capabilities of large language models for general retrieval rewriting. (2) Hyde (Gao et al., 2023a) employs zero-shot prompting to guide large language models in creating a pseudo-document that captures the semantics of the target document, which is then used for retrieval. For fine-tuning methods: (1) Rewrite-Retrieve-Read (RRR) (Ma et al., 2023) uses the accuracy of model responses obtained through rewriting and retrieval as a reward signal to fine-tune the T5 model via reinforcement learning. (2) RQ-RAG (Chan et al., 2024) constructs a dataset of search queries across multiple scenarios and trains a model to perform rewriting, decomposition, and clarification of the original queries. Additionally, we compare the effectiveness of using the original query directly, without rewriting, referred to as Original Query Retrieval (OQR).

### 3.1.4 IMPLEMENTATION DETAILS

The experiments are implemented using PyTorch and employ several LLMs for our rewriting tasks, demonstrating that our DMQR-RAG framework is applicable to various models, including Llama3-8B (Dubey et al., 2024), Qwen2-7B Yang et al. (2024), and GPT-4 (Achiam et al., 2023). After generating multiple rewrites, document retrieval and response generation are conducted according to the setup outlined in Section 2.1. Specifically, each proposed rewriting method independently retrieves 10 documents, recalling a total of 50 documents, which are then reranked by the reranker. By default, we use the Bing search engine as the retriever and BGE (Chen et al., 2009) as the reranker. Both the baselines and our method utilize GPT-4 as the response model, leveraging the reranked Top-5 documents as additional context.

### 3.2 RESULTS

### 3.2.1 BASELINE COMPARISON

The main results are shown in Table 2. Overall, our method outperforms others in most scenarios. The detailed analysis leads to the following conclusions.

**The original query can be effective.** The performance of some rewriting methods (e.g., RRR, Rewrite, and Hyde) is inferior to that of original query retrieval (OQR) in certain scenarios. This suggests that the original query can sometimes accurately express users' intentions and provide

Table 3: The generalization testing results. Our DMQR-RAG can be effectively applied to much smaller LLMs (e.g., Llama3-8B and Qwen2-7B) than GPT-4.

| Method | AmbigNQ | | | | HotpotQA | | | | FreshQA | | |
|---|---|---|---|---|---|---|---|---|---|---|---|
| | H@5 | P@5 | EM | F1 | H@5 | P@5 | EM | F1 | H@5 | P@5 | Acc |
| Llama3 + Rewrite | 77.64 | 44.99 | 49.87 | 62.17 | 39.36 | 17.82 | 35.27 | 46.83 | 71.00 | 45.97 | 71.50 |
| Llama3 + Hyde | 64.80 | 42.41 | 42.88 | 53.77 | 34.47 | 17.44 | 31.82 | 42.23 | 53.83 | 34.61 | 59.17 |
| Llama3 + Ours | **86.04** | **58.69** | **54.75** | **67.53** | **51.22** | **26.23** | **39.65** | **52.51** | **73.33** | **57.30** | **74.50** |
| Qwen2 + Rewrite | 78.54 | 45.34 | 50.31 | 62.35 | 41.54 | 18.83 | 35.50 | 47.25 | 71.02 | 43.35 | 71.19 |
| Qwen2 + Hyde | 59.96 | 36.63 | 39.42 | 50.02 | 31.24 | 15.13 | 31.23 | 41.63 | 53.77 | 34.21 | 59.13 |
| Qwen2 + Ours | **86.23** | **58.25** | **54.78** | **67.37** | **50.71** | **25.77** | **39.88** | **52.49** | **76.38** | **58.69** | **75.04** |
| GPT-4 + Rewrite | 81.06 | 47.17 | 51.24 | 63.96 | 42.08 | 18.91 | 35.47 | 47.39 | 70.67 | 44.61 | 70.83 |
| GPT-4 + Hyde | 79.65 | 59.36 | 53.95 | 64.83 | 46.92 | 25.23 | 39.36 | 51.74 | 61.10 | 44.04 | 62.83 |
| GPT-4 + Ours | **88.08** | **62.43** | **55.24** | **68.57** | **54.18** | **27.93** | **41.12** | **53.99** | **77.83** | **60.03** | **76.67** |

valuable context that enhances document retrieval and end-to-end responses. This, in turn, validates our design of including both the original query and its rewritten versions in the strategy pool.

**Multi-query rewriting is generally better than single-query rewriting.** For document retrieval, our DMQR-RAG outperforms existing rewriting methods across all datasets. Notably, compared to the best baseline, DMQR-RAG shows a significant improvement in P@5 of 14.46% in FreshQA. Moreover, our method achieves substantial improvements in the complex multi-hop questions of HotpotQA, with increases of approximately 8%. This indicates that our method performs well across various types of queries, demonstrating its versatility. For end-to-end response, DMQR-RAG surpasses the best baseline, Hyde, on the AmbigNQ dataset, achieving 1.30% and 3.74% higher EM and F1 scores, respectively. On the FreshQA dataset, it exceeds Rewrite by 5.84% in accuracy. This shows that the documents retrieved by our rewriting method provide the response model with accurate external knowledge, significantly enhancing its response performance. However, as RQ-RAG is specifically designed for solving complex multi-hop questions, it achieves the best results on the HotpotQA dataset. Nevertheless, our method yields competitive results across various types of queries, showcasing its generality.

**Our DMQR-RAG surpasses vanilla RAG-Fusion.** Both DMQR-RAG and RAG-Fusion demonstrate strong performance across the three academic datasets. However, our information-based multi-query approach often achieves slightly better results than RAG-Fusion, particularly on the AmbigNQ dataset, where it shows approximately a 10% improvement in P@5. Furthermore, we will demonstrate that DMQR-RAG can significantly outperform RAG-Fusion through adaptive rewriting selection in more challenging scenarios.

### 3.2.2 GENERALIZATION TESTING

One important question is whether DMQR-RAG is applicable to other LLMs, particularly smaller models than GPT-4. To address this, we tested Llama3-8B and Qwen2-7B, with the results presented in Table 3. The findings indicate that our method is not restricted to high-performing LLMs like GPT-4; in fact, it can be effectively applied to both Llama3-8B and Qwen2-7B, yielding strong results than baseline rewriting methods. This highlights the versatility and generalizability of our approach across different model architectures.

### 3.2.3 ABLATION STUDY

We conduct an ablation study to investigate the effectiveness of each rewriting method. Specifically, we use Llama3-8B as the base model and remove each method individually, comparing the results with our comprehensive multi-query rewriting approach. The results are presented in Table 4.

**Different rewriting methods have varying impacts.** The PAR method contributes the most compared to other approaches, with an average decrease of 1.34% measured by P@5. This suggests that

Table 4: The results for ablation study. The best results are in bold, and the second-best results are underlined. The gray color indicates the worst results.

| Method | AmbigNQ | | | | HotpotQA | | | | FreshQA | | | Avg(↓) |
| | H@5 | P@5 | EM | F1 | H@5 | P@5 | EM | F1 | H@5 | P@5 | Acc | P@5 |
|---|---|---|---|---|---|---|---|---|---|---|---|---|
| Ours | **86.04** | **58.69** | **54.75** | **67.53** | **51.22** | **26.23** | **39.65** | **52.51** | 73.33 | **57.30** | 74.50 | - |
| w/o GQR | 85.65 | 58.02 | 54.04 | 67.26 | 51.18 | 25.81 | 39.37 | 52.00 | 74.00 | 56.60 | **75.33** | 0.60 |
| w/o KWR | 85.50 | 57.67 | 54.37 | 67.40 | 50.64 | 25.47 | 39.37 | 52.00 | 73.67 | 56.40 | 72.50 | 0.90 |
| w/o PAR | 85.48 | 56.22 | 53.69 | 66.84 | 50.03 | 25.24 | 38.98 | 51.58 | **74.17** | 56.73 | 74.67 | **1.34** |
| w/o CCE | 85.58 | 57.96 | 54.38 | 67.35 | 50.47 | 25.58 | 39.57 | 52.21 | **74.17** | 56.77 | 74.00 | 0.64 |

the generated pseudo-answers differ significantly from other rewritings, thereby expanding the information available and further enhancing retrieval diversity. In contrast, the GQR method has the least impact. This is primarily because, in most cases, the rewriting results of GQR are not significantly semantically different from the original queries, especially when the queries are relatively clear. Nevertheless, this does not imply that we should disregard this approach, as user queries in practical scenarios often contain substantial noise.

**Each rewriting method contributes positively on average, but using all methods together is not optimal for specific datasets.** Evidence shows that removing any rewriting method results in an average decrease in P@5, indicating that each method contributes positively overall. However, on the FreshQA dataset, removing any single method could potentially enhance performance, particularly reflected in the increase of the H@5 and Acc metrics. This suggests that using all methods together may introduce more irrelevant documents (i.e., noise) than relevant ones. *Therefore, adaptively selecting the appropriate rewriting methods for each dataset, or even for each query, is crucial for minimizing noise and improving performance.*

### 3.2.4 EVALUATION OF ADAPTIVE REWRITING SELECTION

The previous section shows that removing a rewriting method may lead to performance improvements on the FreshQA dataset when using Llama3-8B as the rewriter. In this section, we validate our adaptive rewriting selection method proposed in Section 2.2.2 with both Llama3-8B and GPT-4 as the rewriters, focusing on the FreshQA dataset. The results are presented in Figure 2 and 3.

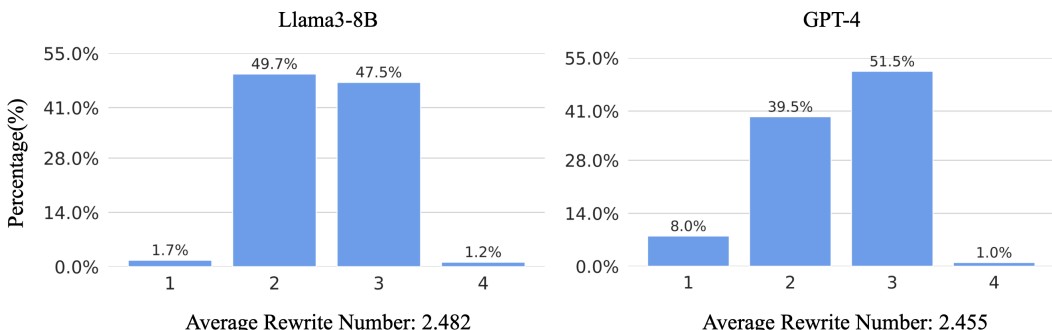

Figure 2: The results of adaptive rewriting selection: distribution of rewriting number.

From the perspective of the number of rewrites (i.e., Figure 2), two interesting conclusions can be drawn. (1) The average number of rewrites after dynamic selection is significantly lower than the original four rewrites, with Llama3-8B and GPT-4 averaging 2.482 and 2.455 rewrites, respectively (a reduction of nearly 40%). (2) The distribution of the number of rewrites follows a Gaussian pattern, which indirectly validates the effectiveness of our method. Specifically, too few or too many rewrites (i.e., 1 rewrite or 4 rewrites) can retrieve insufficient relevant documents or excessive noise, both of which are detrimental to subsequent results.

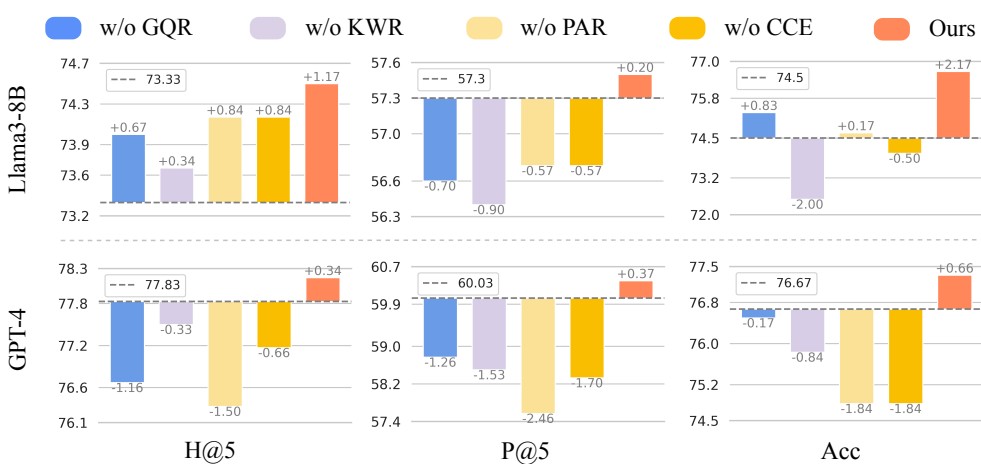

Figure 3: The results of adaptive rewriting selection: retrieval and answer performance.

From the perspective of the retrieval and answer performance (i.e., Figure 3), two additional interesting conclusions can be drawn. (1) Dynamic selection consistently improves these performance metrics, likely due to the reduction of irrelevant noisy documents by avoiding unnecessary rewrites. This demonstrates the effectiveness of our method in dynamically selecting the appropriate rewriting strategy based on the characteristics of the query. (2) The performance improvement of Llama3-8B is larger than that of GPT-4 (e.g., the Acc increases by 2.17 and 0.66 for Llama3-8B and GPT-4, respectively). This difference may be attributed to Llama3-8B's relative lack of power, which results in more redundant rewrites in its original set of four. Consequently, the reduction of these redundant rewrites leads to a more significant improvement.

Overall, adaptive selection enhances performance with fewer rewrites for both the closed-source GPT-4 and the open-source Llama3-8B, demonstrating its broad applicability.

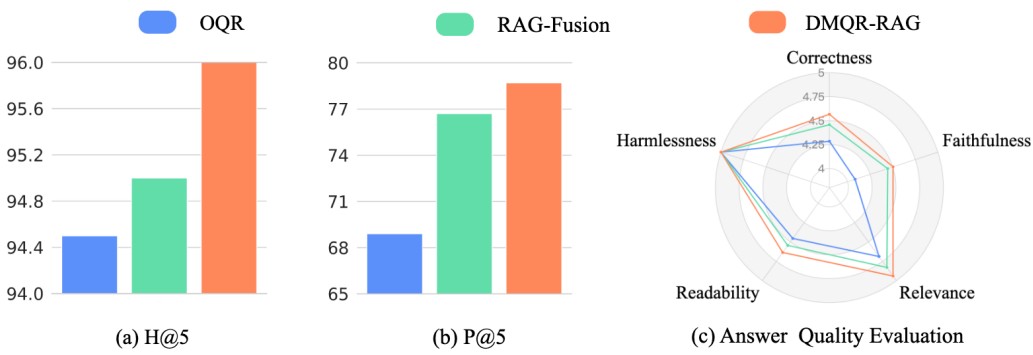

Figure 4: The results from real-world industry scenarios.

### 3.2.5 INDUSTRIAL APPLICATIONS

We deploy the DMQR-RAG framework in real-world industrial scenarios, using queries from 15 million online users. The queries include news-related topics, complex knowledge-based questions, and daily conversational queries. We compare our method with OQR and RAG-Fusion, maintaining consistency with prior experimental settings. Results shown in Figure 4 indicate that our method significantly improves retrieval, with H@5 increasing by an average of 2.0% and P@5 by 10.0%. In terms of end-to-end response performance, Correctness has improved [4], demonstrating that our

---

[4] Note that even a small increase can indicate a significant proportion of problems being solved. For further details, please see Appendix B.

method effectively addresses more user queries. Relevance has also increased, suggesting that while recalling more useful information, our method reduces noise in the retrieved documents. Moreover, the improvements in certain metrics do not come at the expense of others.

### 3.2.6 CASE ANALYSIS

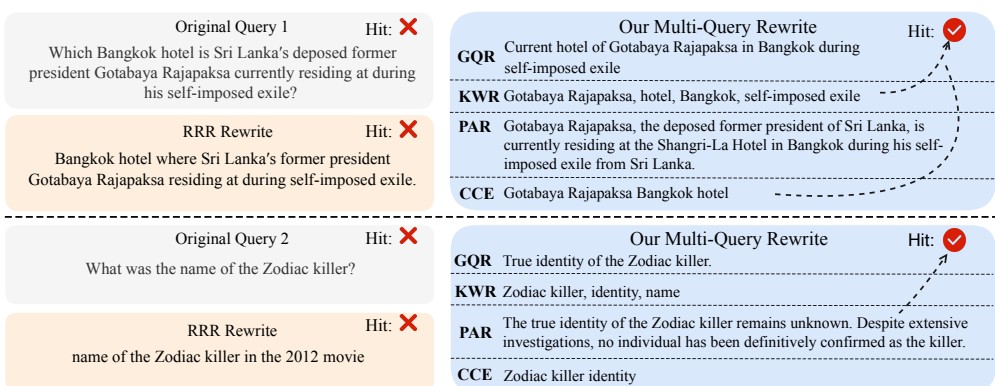

Figure 5: The case studies for complex and simple queries, respectively.

Figure 5 provides a detailed analysis to enhance understanding of our methods. For complex and lengthy queries, the RRR rewrites, despite removing irrelevant information, still produced queries that were too complicated for effective retrieval. In contrast, our approach's keyword rewriting (KWR) and core extraction (CCE) simplified the queries while retaining essential elements, such as "Gotabaya Rajapaksa Bangkok hotel", successfully retrieving the correct documents. For concise queries, pseudo-answer rewriting (PAR) generated content that was semantically closer to the correct answer, such as "remains unknown" and "no individual has been definitively confirmed". Consequently, the pseudo-answers led to successful retrievals, while other rewrites fell short. In summary, the analysis shows that the four rewrites generated by our approach exhibit excellent diversity, each with unique characteristics, making them complementary for different types of queries.

## 4 CONCLUSION

This paper presented the Diverse Multi-Query Rewriting Framework (DMQR-RAG), aimed at enhancing both document retrieval and final responses in retrieval-augmented generation. We developed four rewriting strategies based on information levels to ensure that the rewritten queries are diverse and provide unique insights. Additionally, we implemented an adaptive rewriting selection method utilizing lightweight prompting and few-shot learning. Our evaluation on both academic and industry datasets demonstrated that multi-query rewriting generally outperforms single-query rewriting, with DMQR-RAG surpassing vanilla RAG-Fusion. Our ablation study and case analysis further highlighted the importance of query-specific rewriting strategy selection, confirming the effectiveness of our approach. In the future, we will explore further enhancements to the DMQR-RAG framework, including optimizing the adaptive rewriting selection method and expanding the range of rewriting strategies to create a more diverse strategy pool.

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

## A    RELATED WORK

Existing rewriting methods can be categorized into two categories: training-based and prompt-based. Training-based methods (Wang et al., 2024a; Mao et al., 2024; Ma et al., 2023) use labeled data for supervised fine-tuning on the model or employ reward scores for reinforcement learning to achieve better rewriting results. RQ-RAG (Chan et al., 2024) constructs an innovative dataset that contains search queries and rewritten results across multiple scenarios, which is used to train an end-to-end model that refines search queries. RRR (Ma et al., 2023) proposes a novel training strategy for rewriting, which leverages the performance of response model as a reward and optimizes retrieval queries through reinforcement learning. However, these methods require substantial costs for dataset construction and training.

Prompt-based methods (Zheng et al., 2024; Chan et al., 2024; Wang et al., 2023a) leverage various prompting strategies to directly instruct large models to perform multiple rewriting tasks. Hyde (Gao et al., 2023a) utilizes LLMs to generate a pseudo-answer for the original query in advance. This pseudo-answer is semantically closer to the correct answer, making it easier to retrieve the correct results. Step-back Prompting (Zheng et al., 2024) tackles queries with extensive details by rewriting them at a higher conceptual level to retrieve more comprehensive answers. Least-to-most prompting (Zhou et al., 2023) decomposes a complex query into several easier-to-address subqueries, which are individually retrieved to gather all documents necessary to answer the original query. While avoiding additional training costs, these methods focus only on specific query types, lack generalizability, and produce retrieval results with insufficient diversity. Therefore, our approach proposes multi-strategy rewriting, using prompt-based methods to guide the model in preforming multiple rewrites according to different strategies. This effectively addresses various types of queries and enhances the diversity of retrieval results.

## B    THE EVALUATION CRITERIA IN INDUSTRIAL SCENARIO

We use two anonymous internal models of 13B and 7B sizes as the rewriter and generator, respectively. To measure the performance of end-to-end responses, we incorporated several relevant metrics: Correctness, Faithfulness, Relevance, Thoroughness, Harmlessness, Readability, Logicality, and Creativity, each rated from 0 to 5.

**Correctness**    The correctness criterion measures the accuracy of the response in relation to the initial query. It assesses whether the provided answer is factually accurate and directly addresses the question posed. Correctness is fundamental as it builds the trustworthiness of the system in an industrial scenario, where precision is paramount.

**Faithfulness**    Faithfulness refers to the accuracy and factual consistency of the generated response with respect to the source information from which it is derived. An answer is considered faithful if the claims made in the answer can be inferred from the context.

**Relevance**    The concept of answer relevance entails that the response should address the asked question without introducing other useless information. An answer is deemed relevant if it properly tackles the question. This suggests that the metric does not consider the factuality of the response, but it does incur penalties if the answer is either incomplete or contains superfluous information.

**Thoroughness**    The thoroughness criteria focus on how adequately the response solve the problem. An answer is considered detailed when it not only answers the question but also elaborates on the how and why, encompasses related dimensions, and anticipates follow-up questions or concerns that might arise from the initial inquiry

**Harmlessness**    The criterion of harmlessness emphasizes the importance of generating responses that do not perpetuate or incite harm or bias. This includes avoiding language that could be considered offensive, discriminatory, or incendiary. It also encompasses ensuring that responses do not propagate misinformation or dubious claims that could lead to real-world consequences.

**Readablity**   The readability criterion assesses how easily the text of the response can be read and understood by users. A highly readable response is one that uses clear language, avoids unnecessary jargon, and structures information in a way that is consistent and easy to follow

**Logicality**   In scenarios involving reasoning tasks, the logicality criterion is particularly crucial. It ensures that responses are not only accurate but also logically structured, providing clear and rational explanations for how conclusions are derived from the presented facts. This is vital for maintaining the system's reliability and user trust.

**Creativity**   This criterion evaluates the novelty and uniqueness of the responses provided to a query. It ensures that the solutions are not only effective but also innovative, potentially offering new perspectives or methods that may improve upon existing processes.

## C   PROMPT FOR FOUR QUERY REWRITING METHOD

Table 5: The prompt for four rewrite generation.

### Instruction ###
You will receive a user's question that requires retrieving relevant content through internet search to provide an answer. There are now the following four rewriting methods, General Search Rewriting, Keyword Rewriting, Pseudo-Answer Rewriting, Core Content Extraction. Please apply four rewriting methods to rewrite the question.

### General Search Rewriting ###
Rewrite the question into a general query for internet search.

### Keyword Rewriting ###
Extract all keywords from the question and separate them with commas, preserving the amount of information as in the original question.

### Pseudo-Answer Rewriting ###
Generate an answer for the question, and use the answer to match the real answers from the search engine.

### Core Content Extraction ###
Reduce the amount of information in the original question, only extracting the most core content. The rewritten query should be more brief than Keyword Rewriting.

### Example ###
Question: Which city was the site where the armistice agreement officially ending World War I was signed?
Output:
General Search Rewriting: City where World War I armistice agreement was signed
Keyword Rewriting: World War I, Armistice, Signing Location
Pseudo-Answer Rewriting: The armistice that ended World War I was signed in the city of Compiègne.
Core Content Extraction: World War I armistice signing city

Begin! Only output the final result without any additional content. Do not generate any other unrelated content.
Question: {query}
Output:

