# OpenReview forum: "DMQR-RAG: Diverse Multi-Query Rewriting in Retrieval-Augmented Generation"
_ICLR.cc/2025/Conference — ICLR 2025 Conference Withdrawn Submission_

### Official Review · Reviewer_6nPT · 2024-10-29

**Soundness:** 3
**Presentation:** 3
**Contribution:** 2
**Rating:** 5
**Confidence:** 4

**Summary:**

This paper introduces DMQR-RAG, a Diverse Multi-Query Rewriting framework aimed at enhancing document retrieval and response quality in Retrieval-Augmented Generation (RAG). The authors also present a rewriting strategy selection method that identifies the most suitable approach for each query, thereby improving performance with fewer rewrites. The framework has demonstrated robust performance in both academic and industrial settings.

**Strengths:**

1. DMQR-RAG has shown effective results.
2. Extensive case studies and ablation experiments were conducted, and the framework was deployed in real-world industrial scenarios.
Weaknesses:

**Weaknesses:**

1. The technical depth of the four rewriting strategies and the adaptive rewriting selection method is insufficient.
2. The paper appears to contribute primarily a prompt without additional significant contributions.
3. The adaptive rewriting strategy selection significantly increases the number of tokens and inference latency, which are not addressed in the paper. This is due to DMQR-RAG requiring the input of recall results from multiple queries into the LLM's prompt.

**Questions:**

See Weaknesses

---

> ### Author Response · Authors · 2024-11-22
> **To Reviewer 6nPT**
>
> Thank you for the feedback. Our response is detailed below.
> > Q1: The technical depth of the four rewriting strategies and the adaptive rewriting selection method is insufficient.
> Q2: The paper appears to contribute primarily a prompt without additional significant contributions
>
> The primary focus of our work is on developing a robust ensemble strategy and verifying its feasibility. Therefore, experimental validation is our main approach. We conducted extensive experiments and also tested our strategy in industrial scenarios, all of which confirmed the effectiveness of our strategy.`Please find our response in general comments to all reviewers and AC`.
>
> > Q3: The adaptive rewriting strategy selection significantly increases the number of tokens and inference latency, which are not addressed in the paper.
>
> Regarding cost analysis, the only distinction between our proposed method and the traditional RAG process lies in the query rewrite phase,`as detailed in the fifth point of our summary in general comments to AC and all reviewers`.

---

> > ### Comment · Reviewer_6nPT · 2024-11-25
> >
> > Thank you for your reply. However, from the perspective of a information retrieval researcher, the actual technical contribution of this paper to the RAG community is still very limited

---

### Official Review · Reviewer_hPgY · 2024-11-04

**Soundness:** 2
**Presentation:** 2
**Contribution:** 1
**Rating:** 3
**Confidence:** 4

**Summary:**

This paper introduces a strategy for query rewriting, aimed to enhance RAG systems' performance. In particular, this strategy would encourage rewriting the user query into multiple diverse queries by combining 4 different query rewriting strategies. This paper uses a fixed retriever + reranker setting where the retriever is bing search engine and the reranker is BGE-reranker. On three test datasets, this strategy developed by this paper (named DMQR) is comparable (maybe slightly better) to another previously published multi-query rewriting strategy (RAG-Fusion), and outperforms other baselines. This paper further introduce another prompt that allows a subset of query rewriting strategies to be selected based on the specific data point. This adaptive strategy is designed to reduce the amount of distracting documents during retrieval. The paper showed that in some industrial-setting datasets, adaptive DMQR proposed by the paper slightly outperforms RAg-Fusion.

**Strengths:**

The paper introduces a strategy for rewriting user queries into multiple diverse queries. These diver queries are aimed to retrieve more diverse and comprehensive documents, which would later be used for a generator such as GPT4. In order to reduce the irrelevant documents retrieved by some non-satisfactory rewrites, the paper proposes an adaptive strategy which is a specialized prompt that instruct an LLM to select only the appropriate rewriting strategies based on a specific user query, so data-specific. One of the paper's main strength is that it uses both commercial and open-sourced language model generator such as GPT4 and Llama3-8B, which shows their adaptive DMQR RAG is generalizable to different LLMs as generator. The paper also conduct comprehensive ablation studies to investigate the individual effect of each of the rewriting strategies used in the DMQR. Lastly, the paper was able to show that adaptive DMQR outperform RAG Fusion in industrial setting.

**Weaknesses:**

1. Lack of originality. I think the idea of combining multiple rewritten queries instead of a single rewritten query has already been proposed in a previous work RAG-Fusion. Therefore, this work is not the first to propose using multiple rewritten queries to improve RAG performance. The authors claim their rewritten strategies are "based on different levels of information", which is vague. In fact, all strategies used in authors' proposed DMQR: removing noise and excessive details from user queries, expanding user queries (i.e. query expansion), and keyword extraction, are fairly natural ideas and many of which I am sure have been explored by previous work before, such as query expansion. This paper fails to engage in a meaningful discussion of of previous work. While it cites a good number of papers in section 2.2.1, it does not discuss what these papers do and how the authors proposed ideas, information equality, expansion and reduction are different from these previous papers. I highly encourage the authors to include a formal related work section. Furthermore, all ideas expressed in this paper, from query rewrite strategies to adaptive selection, are all accomplished by prompt engineering. Prompt engineering itself is not a source of adequate innovation and originality. As it stands right now, I fail to see the true distinctions of this work and previous works, especially RAG Fusion. I would rate the originality of this work to be low, and I am inclined to reject this paper primarily because of this reason.

2. The paper's writing is very unclear, it lacks a related work section that properly acknowledge previous work and explain why the current work is different. It talks about "transmission of information from queries to documents" in the introduction, and never explained the meaning of the sentence. There are too many acronyms which prevents easy understanding. I also believe section 2.1 is completely not necessary.

3. Another concern is that this work only uses bing + BGE-reranker as the sole retriever. While the paper claims this is its advantage, I think it is completely nonsensical to only use one retriever. I also don't understand the claim "without loss of generality", why is this specific bing+ BGE retrieval pipeline can be considered as "without loss of generality"? Later on in the paper, the authors even used different LLMs as generator, showing generalizability of their approach. So why do the authors not wishing to demonstrate generalizability across different retrieval settings? There are many unknown systematic bias introduced with a fixed (and not generalizable) retrieval setting. I would encourage the authors to apply their DMQR strategy on other retrieval settings, including using state-of-the-art semantic embeddings (OpenAI-v3, Grit-LM) and alternative retrieval strategies such as RankGPT and RankLLama. This is one of the major weakness of the paper, failing to show generalizability across retrieval settings.

4. The paper does not provide bootstrapped confidence intervals and p-values, both of which are necessary to examine with statistical rigor if one approach is better than another approach. Another main concern is this, the authors specifically mentioned in their paper that Table 2 shows DMQR is better than RAG-Fusion w.r.t P@5 metric, this feels like selective reporting.

**Questions:**

1. For all tables, especially table 2, I would suggest authors to bold numbers that are statistically different from the second-best number, otherwise please bold both numbers to avoid confusion. Figure 4 should include error bars, and Figure 4 (c) should include some form of indications for p-value, confidence-level or error bars.

2. I would like to see a proper discussion that clearly describes the difference of the current paper's DMQR strategy, and other works such as RAG Fusion. I would also like the authors to discuss the four ideas proposed in the paper, general query rewrite, keyword extraction and rewrite, query expansion, query reduction,  and how they relate or differ from previous work. I know for a fact that there are many works on query expansion, so at least that idea is not novel. As I explained before, the lack of proper related work discussion prevents me from seeing the true originality (if there is any) of this paper.

3. I think limiting the setting to only bing + BGE is a potential bias-inducing practice. I would like to see authors compare their method with RAG Fusion while using different retrievers, as mentioned before.

If my concerns are addressed, I would adjust my scores accordingly.

---

> ### Author Response · Authors · 2024-11-22
> **To Reviewer hPgY**
>
> We sincerely appreciate your valuable suggestions for improving our method. Your concerns are addressed below.
> > Q1: Suggestions for improving statistical clarity in tables and figures
>
> To address your feedback on our manuscript, while we appreciate your suggestion regarding bolding numbers and including additional elements in the figures, we believe that the current format sufficiently conveys the necessary information. To explain further:
> 1. Concerning bolding numbers that are statistically distinct from the second-best results in our tables: We have opted for a uniform formatting style across the document to ensure simplicity and ease of reading.
> 2. As for adding error bars and indicating p-values or confidence levels in Figure 4, specifically in Figure 4(c): We aimed to keep the figures as clear and straightforward as possible, directing the readers to the text for the detailed statistical context. However, adding p-values, confidence levels, or error bars indeed enhances the rigor of the experiments. We will consider incorporating these elements in subsequent experiments.
>
> > Q2: I would like to see a proper discussion that clearly describes the difference of the current paper's DMQR strategy, and other works such as RAG Fusion.
>
> In fact, `as detailed in the first and the second point of our summary in general comments to AC and all reviewers`, our focus lies on introducing a novel ensemble strategy, rather than a new query rewrite method. Nonetheless, we have also introduced two new rewriting techniques: keyword rewriting and core content extraction.
>
> > Q3: I think limiting the setting to only bing + BGE is a potential bias-inducing practice. I would like to see authors compare their method with RAG Fusion while using different retrievers, as mentioned before.
>
> Regarding your suggestion to compare different retrieval systems, we fully understand the importance of this for enhancing the reliability and generalizability of the research. However, due to the scope and focus of our current study, we have chosen to restrict our experiments to the Bing + BGE setup. This decision is primarily based on the following considerations:
>
> 1. Our primary interest lies in assessing whether different LLMs used as rewriters can improve the results of retrieval and generation. To better analyze the quality of the rewriting and to avoid introducing additional variables, it is necessary to keep the retrieval, reranking, and generation modules consistent. This approach helps to prevent the outcome from being skewed by performance variations in other modules.
> 2. The Bing + BGE combination, widely recognized as a robust configuration for retrieval and reranking, provides a solid baseline that facilitates direct comparisons with other research findings.
>
> Despite this, we acknowledge that experiments conducted under different retrieval system configurations possess unique value. In our future work, we will explore how methods perform under various retrieval configurations.

---

> > ### Comment · Reviewer_hPgY · 2024-11-25
> >
> > Thank you for your response. Showing robustness of your approach across a variety of setting is important. In your future work, consider using other retrievers and other rerankers. There should also a sensitivity analysis on how your approach performs when the user query is already high-quality, for example, scientific queries. Since the current version of the paper does not show that, I will keep my score as it is.

---

### Official Review · Reviewer_82A3 · 2024-11-05

**Soundness:** 2
**Presentation:** 3
**Contribution:** 2
**Rating:** 3
**Confidence:** 2

**Summary:**

This paper proposes an approach to create diverse rewrites for RQG. The assumption is that different rewrites may account for different types of queries. The rewrites are obtained through different prompts to an LLM. Another phase of adaptive selection strategy is used to select the generated rewrites via a few-short prompt. The experiments performed on 3 datasets show that this approach can outperform the original query and the single rewrites created by other methods. It also slightly outperform the alternative fusion method of different RAGs.

**Strengths:**

1. The assumption that different rewrites can reflect different aspects of queries is reasonable. The creation of diversified rewrites has not been widely explored in the RAG literature.
2. The strategies used in this paper - prompting an LLM - for different types of rewrites is also reasonable. This is a commonly used strategy in literature.
3. The experiments show interesting results. Although it only outperforms slightly the fusion method, the study shows that creating diverse rewrites is a reasonable alternative to it.
4. The ablation analysis shows the impact of different types of rewrite, as well as of different LLM used.

**Weaknesses:**

1. The proposed approach is based on a range of the existing ones. The only originality lies in a combination of previous rewrites. The  basic idea is quite similar to the fusion method. It is unclear if conceptually, the proposed method is better than the fusion method (which is still quite simplistic). Overall, the novelty of the approach is limited.
2. In addition, everything is done through prompting. Despite the fact that some improvements have been obtained, it is difficult to understand how the diverse rewrites created have been incorporated in the selection phase. The paper mentions several interesting targeted problems with the single rewrites (noise, etc.), the problems are not further analyzed later. It is simplistic to assume that using relevant prompts would solve the problems. It would be great if these problems are further analyzed in the experiments.
Despite the reasonable assumption made at the beginning of the paper, one can hardly see how the problem has been solved by the two steps of prompting. Therefore, the proposed approach and experiments only provide a quite superficial understanding of the problem.
3. When multiple rewrites are generated through multiple prompts, there is obvious an additional cost. This aspect should be discussed.
4. In the baselines, the prompts are zero-shot prompts, while in the proposed one selection phase, few-shot learning is used. This may give some advantage to the proposed method. It may be fairer to try to use few-shot learning in the baselines.

**Questions:**

1. How have the proposed method solve the different problems mentioned in the paper? Does it reduce noise? For what queries each type of rewrite is the most useful? Instead of asking an LLM to make the selection, would it be better to train a small selector for it?
2. Have you compared with more sophisticated fusion methods? There are a number of alternative approaches proposed in the IR literature for result fusion. They could be considered as alternatives.
3. The proposed method is related to search result diversification, for which some studies proposed to diversify the search queries. Can the  proposed method be related to these diversification methods?

---

> ### Author Response · Authors · 2024-11-22
> **To Reviewer 82A3**
>
> We thank the reviewer for the encouraging feedback. Regarding the points you raised, we would like to clarify the following:
>
> - Our contributions are not limited to the combination of different query rewrite strategies. `Please find our response in general comments to all reviewers and AC, specifically the contribution part`.
> - Regarding cost analysis, the only difference between our proposed method and the traditional RAG process lies in the query rewrite phase, `as detailed in the fifth point of our summary in general comments to AC and all reviewers`.
> - When analyzing baselines experimentally, we adhered to the original settings proposed by the developers of each method. If the proponents of a method employed a few-shot approach, we conducted our experiments using a similar few-shot setup. For those methods that do not use few-shot, such as rag-fusion, we have tried adding few-shot, and the results show it is on par with or even inferior to zero-shot.
> ---
> > Q1. 1: How have the proposed method solve the different problems mentioned in the paper? Does it reduce noise?
>
> Thank you for pointing these out. Some experiments in the paper manage to illustrate certain facts.
>
> In the ablation study section, we validated that each rewriting strategy plays a distinct and crucial role in complementing the weaknesses of other rewriting strategies. Additionally, in Section 3.2.6, we conducted a case study to demonstrate that our method effectively addresses the following issues:
>
> 1. Keyword Rewriting aligns with search engine preferences, thereby recalling more useful documents.
> 2. Core Content Extraction discards some detailed content to perform retrieval at a coarser granularity, which in turn recalls more relevant documents.
> 3. Pseudo-Answer Rewriting matches the pattern of answers, thereby hitting more relevant documents.
>
> In the future, we will conduct more experiments to further refine this analysis.
>
> > Q1. 2: For what queries each type of rewrite is the most useful? Instead of asking an LLM to make the selection, would it be better to train a small selector for it?
>
> Each query rewriting method serves a different purpose, as can be seen from the case studies. However, in some instances, it is not necessary to perform rewriting four times; three or even fewer rewrites can also retrieve relevant documents, as evidenced by the ablation studies. Consequently, we have designed a series of guidelines to guide the LLM in making selections, `which shows in the third point of our summary in general comments to all reviewers and AC`.
>
> We could train a small selector to handle this task. However, considering that the focus of our current work is to validate the concept of an information-based ensemble approach, and not on training a new model, `as detailed in the first point of our summary in the general comments to the AC and all reviewers`, we did not pursue this in the present study. In future work, we plan to explore training a compact selector model.
>
> > Q2: Have you compared with more sophisticated fusion methods? There are a number of alternative approaches proposed in the IR literature for result fusion. They could be considered as alternatives.
>
> We have reviewed fusion methods in the IR field, which primarily involve decomposing queries into sub-queries based on their intent, and then conducting searches for these sub-queries (Wang et al., 2024a; Besta et al., 2024) not to adopt this approach for the following reasons:
>
> 1. This approach addresses problems similar to those tackled by RQ-RAG in our baseline.
> 2. Our emphasis is on generic rewriting. In contexts with multiple intents, our method can be applied to each of the aforementioned sub-queries, allowing for multi-route rewriting and retrieval for each sub-query. The focus of these two approaches is different: one emphasizes the decomposition of intents, while the other emphasizes multi-route rewriting for each intent/query.
> Our experimental results on the HotpotQA dataset demonstrate that our method is still capable of effectively handling complex intent queries.
>
> > Q3: The proposed method is related to search result diversification, for which some studies proposed to diversify the search queries. Can the proposed method be related to these diversification methods
>
> Our baseline can be viewed as a method to diversify search results. The connections between our proposed method and these approaches have been detailed in Appendix A.
>
> [1] RichRAG: Crafting Rich Responses for Multi-faceted Queries in Retrieval-Augmented Generation
>
> [2] Multi-Head RAG: Solving Multi-Aspect Problems with LLMs

---

### Official Review · Reviewer_nxfS · 2024-11-07

**Soundness:** 2
**Presentation:** 2
**Contribution:** 2
**Rating:** 3
**Confidence:** 4

**Summary:**

The paper introduces DMQR-RAG, a novel multi-query rewriting approach aimed at improving document retrieval diversity and recall within the Retrieval-Augmented Generation (RAG) framework. Unlike prior methods, DMQR-RAG leverages multiple rewriting strategies to ensure diverse document retrieval while maintaining high relevance. This approach aims to refine the transmission of information from queries to documents.

While the proposed method shows promise and demonstrates performance improvements across both academic and industry settings, the paper lacks sufficient technical depth and detail, especially regarding the specifics of the rewriting method. Important methodological details are missing, making it difficult to figure out if the hypotheses about improved retrieval and response generation are fully substantiated in the methodology. This lack of clarity around the technical implementation may hinder reproducibility and raise questions about the robustness of the reported improvements. Despite these limitations, the results presented indicate the potential for DMQR-RAG to set a new standard in multi-query rewriting for RAG, though further clarification and technical elaboration would strengthen the work considerably.

**Strengths:**

1) The paper is clear and easy to follow.

2) It addresses an important area in RAG, enhancing retrieval through query modeling.

3) Comprehensive Evaluation: Experiments are thorough, including both academic and industrial settings.

**Weaknesses:**

1.  While Section 2.1 provides an intuitive overview, it lacks specific details on implementing query rewrites. The approach for the three aspects—equality, expansion, and reduction—appears to use a single baseline for each, limiting the exploration of different methodologies. In Section 2.2.2, the selection mechanism determines which rewritings to apply but does not clarify the criteria behind these choices. IMO if only the baselines were used for equality, expansions, and reduction, the technical novelty of this work is limited to asking the LLM to choose between different types of "known" rewriting techniques.

2. The paper lacks clarity on how query rewrites are actually implemented. For instance, the general query rewrite strategy aims to "extract all keywords from the query q, particularly nouns and subjects." However, the exact process for this extraction is unspecified. Thus I don't understand how the handling of extracted elements and the potential noise in proper noun extraction is dealt with. This gap is imprtant, as extraction methods directly impact retrieval performance and can introduce significant variability.

3. The hypotheses in the paper are not well-supported by evidence. For instance, the statement "By enhancing the diversity of information in the rewritten queries, we increase the likelihood of retrieving a broader range of documents, ultimately improving our chances of obtaining genuinely relevant documents" is attributed to general literature (Maron & Kuhns, 1960; Baeza-Yates et al., 1999; Weikum & Vossen, 2001) rather than research directly focused on natural language query diversity. Some references, such as the Weikum and Vossen citation, are database-focused and do not address natural language queries or retrieval diversity directly. Acc. to me this doesn't add to the argument's relevance and rigor. It is also well known that rewriting adds to the drift in query modelling.

4. The related work does not include any of the classical or modern information retrieval literature about query modelling. This also shows in the baselines where only LLM-based query rewriting baselines have been considered.

5. The experiments look comprehensive, but the choice of datasets could have been better argued. Most of the QA datasets are multi hop or complex QA tasks which inherently have multiple aspects. It would be interested to test their effectiveness on simpler QA datasets like natural questions or even ranking datasets like MSMarco or TREC-DL. If DMQR-RAG only works for complex questions, then the authors should position their claims accordingly. In that case they should think about other datasets like strategyQA or WikiMultihopQA.

**Questions:**

1) What are the implementation details of the methods presented in 2.1. Specifically, what novel additions were made to the baselines mentioned in the section explaining equality, expansions, and reduction  ?

2) Did you consider any classical query rewriting methods ?

3) Do your methods work for queries that are not multi hop or are not complex ?

---

> ### Author Response · Authors · 2024-11-22
> **To Reviewer nxfS**
>
> We greatly appreciate your valuable feedback. Firstly, we would like to clarify the following point:
>
> - Regarding the criteria behind the selection mechanism, we have outlined several guidelines provided to the LLM in the form of prompts. We also supplied a few-shot learning examples to enable the LLM to leverage its in-context learning capabilities for autonomous selection. This process `is detailed in the third point of our summary in the general comments to the AC and all reviewers`.
>
> - Regarding why we did not opt for traditional modern query-rewrite strategies, we considered the fact that LLMs possess strong comprehension abilities, fully capable of handling such tasks and surpassing conventional methods. Meanwhile, we chose to maintain consistency with the settings of previous mainstream LLM-based query rewrite papers by selecting an LLM-based approach for comparison. Therefore, we did not consider traditional methods.
> ---
> > Q1: What are the implementation details of the methods presented in 2.1. Specifically, what novel additions were made to the baselines mentioned in the section explaining equality, expansions, and reduction?
>
> Each of our methods is implemented through the prompt engineering. By carefully designing prompts, we accomplish various rewriting tasks. Specific details about the prompts can be found in Appendix C. We conducted experiments with different open-source and proprietary LLMs, demonstrating the versatility of our designed prompts.
>
> Additionally, while our focus remains on how to effectively ensemble different rewriting methods, we have also introduced two new rewriting techniques along with a dynamic selection scheme. `Please refer to our response in the general comments to all reviewers and AC, specifically the contribution part`.
>
> > Q2: Did you consider any classical query rewriting methods ?
>
> We have thoroughly reviewed recent LLM-based query rewriting methods and selected a few representative works from them. Specifically, we have categorized them into the two aspects, as detailed in Section 3 in our paper.
>
> It is worth noting that recent works often focus on proposing a new framework. **For a more effective comparison, we only considered those works purely dedicated to query rewriting**. Additionally, our proposed method is "**plug-and-play**," allowing it to be integrated into any existing framework to further enhance performance.
>
> > Q3: Do your methods work for queries that are not multi hop or are not complex ?
>
> In our experiments, we chose the AmbigQA dataset, a simple and straightforward QA dataset specifically designed to address the inherent ambiguity in Natural Questions. The FreshQA dataset also encompasses both simple and complex QA components. The exemplary performance of our method on these two datasets demonstrates that it can still outperform other rewriting methods in simple QA scenarios.

---

### Author Response · Authors · 2024-11-22
**To AC and All Reviewers**

We thank the reviewers for providing valuable suggestions that helepd us improve our paper.
In response to the feedback, we wish to clarify the following points:

**Contribution**

1. **We propose a general-purpose multi-query rewriting framework** that employs various strategies based on the amount of information. Note that developing and advocating for a new singular rewriting method was not the central objective of our research.
2. **We introduce two innovative rewriting techniques: keyword rewriting and core content extraction** (Although the concept of using keywords for retrieval is prevalent in the industry, our approach marks the first instance of employing a Large Language Model (LLM) to extract keywords).
3. Furthermore, **we innovatively propose a dynamic strategy for selecting rewriting techniques**. By analyzing instances where relevant documents could not be recalled in experiments, we summarized several guidelines. These are provided as prompts to the LLM, accompanied by few-shot examples to leverage the LLM’s capacity for in-context learning, enabling it to autonomously select the most appropriate rewriting method. This is detailed with specific examples in the paper.
> 1. <General Search Rewriting>: If the question is ambiguous or lacks clarity, <General Search Rewriting> may be useful. Conversely, if the question is sufficiently clear, the <General Search Rewriting> is usually not necessary.
> 2. <Keyword Rewriting>: If the question has precise keywords, especially nouns or subjects, which can help search engines locate relevant documents quickly, then the <Keyword Rewriting> may be useful. If the <Keyword Rewriting> rewriting result only simply lists all the keywords or keywords are helpless to the search, like meaningless adverbials and verbs, this method is not suggested. If there is too much keywords in the rewriting result, then <Keyword Rewriting> is also not recommended.
> 3. <Core Content Extraction>: If the question is complex and requires quick identification and extraction of the most crucial information, using <Core Content Extraction> is suitable. Conversely, if the question is short or simple enough, then the <Core Content Extraction> are usually not necessary.
> 4. <Pseudo-Answer Rewriting>: Usually, the <Pseudo-Answer Rewriting> can improve the search quality.
> 5. Make sure that the rewriting method you finally choose does not have almost the same rewriting results.

**Comparison with Other Methods**

4. Compared to rag-fusion, our method emphasizes more targeted and directed multi-query rewriting, rather than simply employing four generic rewrites. It is important to note that the outcome of multiple generic rewrites often appears similar, typically involving synonym replacements, reordering of words, and so on.
5. Regarding cost analysis, **the only difference between our proposed method and the traditional RAG process lies in the query rewrite phase. The retrieval phase is conducted in parallel**, hence the timing remains consistent with that of single-route retrieval. As for the subsequent reranking and response phases, they operate under the same number of documents, resulting in no significant difference in terms of time expenditure.

---

### Note · Authors · 2024-12-16

I have read and agree with the venue's withdrawal policy on behalf of myself and my co-authors.